# Breathing Pattern Analysis in Cattle Using Infrared Thermography and Computer Vision

**DOI:** 10.3390/ani11010207

**Published:** 2021-01-16

**Authors:** Sueun Kim, Yuichi Hidaka

**Affiliations:** Laboratory of Veterinary Surgery, Department of Veterinary Science, University of Miyazaki, 1-1 Gakuen Kibana-dai Nishi, Miyazaki 889-2192, Japan; yhidaka@cc.miyazaki-u.ac.jp

**Keywords:** breathing pattern, infrared thermography, cattle health and welfare, computer vision, machine learning

## Abstract

**Simple Summary:**

Breathing patterns are commonly used to assess cattle health and welfare parameters such as stress, pain, and disease. Infrared thermography has recently been accepted as a non-invasive tool for breathing pattern measurement. In this study, we applied a computer vision method (Mask R-CNN) to infrared thermography and made it possible to automatically estimate the breathing pattern in cattle. Breathing patterns identified by computer vision were highly correlated with those measured through thermal image observation. As this method is not labor-intensive and can handle numerable big data, it might be possible to analyze breathing patterns from various angles in the future.

**Abstract:**

Breathing patterns can be considered a vital sign providing health information. Infrared thermography is used to evaluate breathing patterns because it is non-invasive. Our study used not only sequence temperature data but also RGB images to gain breathing patterns in cattle. Mask R-CNN was used to detect the ROI (region of interest, nose) in the cattle RGB images. Mask segmentation from the ROI detection was applied to the corresponding temperature data. Finally, to visualize the breathing pattern, we calculated the temperature values in the ROI by averaging all temperature values in the ROI. The results in this study show 76% accuracy with Mask R-CNN in detecting cattle noses. With respect to the temperature calculation methods, the averaging method showed the most appropriate breathing pattern compared to other methods (maximum temperature in the ROI and integrating all temperature values in the ROI). Finally, we compared the breathing pattern from the averaging method and that from the thermal image observation and found them to be highly correlated (R^2^ = 0.91). This method is not labor-intensive, can handle big data, and is accurate. In addition, we expect that the characteristics of the method might enable the analysis of temperature data from various angles.

## 1. Introduction

Breathing patterns are a vital sign that can provide valuable information relating to disease [1], stress [2], and pain [3] in cattle. It is common to evaluate breathing with the aid of a stethoscope, but the cattle can be excited through contact and this can affect the breathing patterns. The breathing pattern can also be measured by observing flank movements [4]; however, it can be difficult to observe in cold environments as the movements are typically slower and less pronounced [5]. To overcome these limitations, infrared thermography (IRT) has recently been used to measure the breathing characteristics of cattle [5,6]. In fact, IRT has been used not only to measure breathing but also in various fields such as stress assessment by eye temperature [7,8,9] or nose temperature [10]. However, the previous methods have utilized it in a limited way. For example, they used non-sequence temperature data [7], such as measurements taken three times a day for three days, but this does not provide a temperature pattern. In other cases, the observer monitored the color change around the region of interest (ROI) [5,6] to evaluate the temperature change, which is labor-intensive. Some methods used fixed ROI settings and gained sequence temperature data [8,9,11], but, as we cannot control the animal’s movement, these methods require resetting the ROI or moving the camera in order to maintain the ROI in the desired region.

The recent development of computer vision has broadened the horizons of research, and it has been used in various biomedical fields, including animal pose estimation [12], breathing pattern analysis in humans [13], and head pose estimation [14]. In particular, computer vision was also used for IRT research, where human breathing patterns were analyzed not only with IRT but also with computer vision [13]. They used thermal images and a tracking algorithm, the Median Flow algorithm [15], to automatically detect the ROI (nose) in the thermal images. Then, they made the breathing pattern visible by integrating all temperature values in the ROI, not by averaging them. This was because they detected the ROI (nose) in the thermal image and the detected nose size varied according to the breathing strength. In this case, the average value could not distinguish between deep breathing and shallow breathing.

The application of the same method to cattle has some limitations. First, a better ROI detection algorithm is necessary because the cattle’s movements are too uncontrolled to detect in thermal images. Secondly, infrared studies so far have set the ROI to a square shape, including unnecessary parts other than the nostrils. This effect is significant, especially in cattle, because the distance between the two nostrils is greater than in many other species. Finally, in terms of the temperature calculation, the integrated method could be problematic. The reason for this is that, in the case of cattle, the distance between the two nostrils is great and anatomically three-dimensional, so they are often detected alternately, one nostril or both nostrils. If integration is performed, both detections indicate twice the value of one detection.

To overcome these limitations, we used Mask R-CNN, a state-of-the-art algorithm for object detection. Mask R-CNN is a two-stage detection algorithm which prioritizes detection accuracy over inference speed and has been used in animal welfare evaluation [16,17,18,19] because of its high-quality segmentation mask for each detection. Using Mask R-CNN means we used RGB images, not thermal images, for the ROI detection. RGB images have more clearly defined characteristics than thermal images. In addition, as the algorithm provides a high-quality segmentation mask for each detection, we could set the ROI according to the shape of the nose. Finally, as we used RGB images for the ROI detection, the nose size is not related to the breathing strength. This allowed us to use the thermal value averaging method in the ROI, not the integrating method.

To sum up, in this paper, our purpose is to incorporate computer vision into IRT in breathing pattern analysis in cattle. To assess the computer vision method’s applicability, we compared it with the previously assessed method, thermal image observation [6]. In addition, for the temperature value calculation, we evaluated three different methods: (1) maximum temperature in the ROI, which is the most commonly used method in previous IRT research; (2) integrating all temperature values in the ROI, which was used for computer-vision-based IRT breathing pattern analysis in humans; and (3) averaging all temperature values in the ROI.

## 2. Materials and Methods

### 2.1. Breathing Pattern Analysis through Observation

The temperature distribution on the skin changes with the ambient temperature. These changes could lead to difficulties in identifying the ROI on thermal images because the information about the facial morphology becomes lost. To overcome this problem, the process of translating from a continuous temperature value to its digital color-mapped equivalent, which is called quantization, should be applied. We used the optimal quantization technique [13,20] before observation. This technique dynamically finds the threshold value that best separates objects from background and maximizes the contrast in the thermal image, making it possible to detect subtle signals (Figure 1). After the optimal quantization, the observer monitored the color change that occurred as a result of the thermal fluctuations around the nose during inhalation and exhalation. The time taken to complete five breaths was converted into the number of breaths per minute. In other words, the number of breaths was counted through the color change in the nostril; if it took 10 s to breathe five times, the respiration rate was 30 per minute. This manually calculated respiratory rate was used to evaluate the following algorithm-determined respiratory rate.

### 2.2. Observation through a Computer Vision Method

#### 2.2.1. Object Detection Training

To automatically detect the ROI (nose) in RGB images, it is necessary to train a nose-part annotated dataset. The dataset then includes images and image annotations. We used VIA (the VGG Image Annotator) [21] to annotate the nose part for 1400 images: 1000 images for training and 400 images for validation. For the detection training task, we used Mask R-CNN [22] and transfer learning with a pre-trained model. Transfer learning is the improvement of learning through the transfer of knowledge from a related task that has already been learned [23]. When a dataset is not large enough to train, a model learned from a similar existing dataset is used. Our dataset of 1000 images for training and 400 images for validation was not big enough to train. COCO is a large-scale object detection dataset made by Microsoft [24]. There were parts similar to the COCO dataset [24], which includes animals, so we used a pre-trained backbone weight based on the COCO dataset to accelerate the training process. ResNet 101 [25] was used as the backbone. With ResNet 101 and the COCO dataset, we fine-tuned a new classifier for our nose detection and gained a weight for the detection.

#### 2.2.2. ROI Detection and Temperature Extraction 

For the purposes of the present study, 50 samples comprising RGB images (video) and temperature sequence data were collected from five Japanese Black calves hospitalized in a veterinary teaching hospital (University of Miyazaki, Miyazaki, Japan). We used a mobile thermal imaging camera (FLIR One Pro for iOS (accuracy: ±3.0 °C, sensitivity: 0.07 °C; field of view: 55° × 43°; resolution: 80 × 60; temperature range: −20 to 400 °C), FLIR Systems Inc., Santa Barbara, CA, USA, www.flir.com). FLIR One Pro is a basic entry level camera with some limitations compared to other thermal imaging camera, such as accuracy, sensitivity, battery performance and emissivity setting. However, we used this device considering its portability and commerciality. The temperature sequence data and RGB video were recorded simultaneously using the camera. The temperature sequence data contained all pixel temperature data. For example, if one minute of temperature data were obtained, 184,320,000 temperature values would be included (184,320,000 = 640 (frame height) × 480 (frame width) × 10 frames per second × 60 s). The emissivity was set to 0.95 (the proper emissivity for animal tissue should be 0.98, not 0.95; however, it was set to 0.95 due to the device’s emissivity setting limitation) and the recordings began at 1 m in front of the calf; however, considering its applicability, afterwards, the distance between the camera and the calf was lessened or increased during recording. The RGB images were used to detect the ROI (nose) and gain the mask. After detecting the nose region in the RGB images with the weight obtained from the detection training detailed in Section 2.2.1 (Figure 2), the masks were applied to the thermal temperature data of the corresponding frames and all temperature data were obtained for the ROI (Figure 3). More specifically, in the RGB images, we set the nose part pixels (using the mask) to 1, set the not-nose part pixels to zero, and constructed the matrix (npz file, 640 (frame height) × 480 (frame width) × frame numbers). By multiplying it with the corresponding temperature sequence data, only the temperature values for the exact part of the nostril were obtained. In addition, the average temperature value of the nostril could be obtained by dividing it by the number of pixels corresponding to the mask. We used Python for all these calculations.

### 2.3. Statistical Analysis

#### 2.3.1. mAP

To evaluate the accuracy of the algorithm, the mAP (mean average precision) for object detection was used. AP (average precision) is a popular metric for measuring the accuracy of object detectors. AP is a calculation of the precision and recall based on IoU (intersection over union). IoU measures the overlap between two boundaries: the predicted boundary and the ground truth boundary. We predefined the IoU threshold as 0.5; when the ratio of two bounding boundaries exceeds 50%, precision and recall are calculated. Precision measures the number of predicted trues out of the total number of detections. Recall measures the number of predicted trues out of the number of ground truths. For example, if there is an image with five objects, the algorithm detects three, and the number of accurate object detections is two, then the precision is 66.6% (2/3) and the recall is 40% (2/5). AP is obtained by integrating the precision–recall curve. Finally, the average of the AP becomes the mAP. The average precision value is computed over 0–1. The higher is the value between 0 and 1, the better is the accuracy.

#### 2.3.2. Statistical Analysis of Temperature Calculations and Overall Accuracy

We calculated the temperature data for all pixels of the ROI via three different methods: (1) determining the maximum temperature value in each frame ROI; (2) averaging the temperature values in each frame ROI (after adding the temperature values for all pixels corresponding to the nostril and dividing the sum by the number of pixels corresponding to the mask); and (3) integrating temperature values in each frame ROI (same as the averaging temperature values method but without dividing by the number of pixels corresponding to the mask). Through this process for all frames, we visualized the breathing patterns as temperature changes over time. To decrease ripples in the breathing pattern, we used the Gaussian window (Figure 4). As mentioned above, the time taken to complete five breaths was converted into the number of breaths per minute.

Once the respective breaths per minute was determined via thermal image observation and the computer vision method, a regression analysis was carried out in order to assess the level of agreement between the two types of recording. The temperature value averaging method was selected out of the three different methods for this analysis.

## 3. Results

In the case of human-related object detection, datasets containing large amounts of image data have already been made [24], providing enough data for algorithm training. In this study, we tried to make our dataset to apply computer science to the animal field. It was relatively small compared to the human datasets, and transfer learning was thus applied to maximize efficiency. The training was conducted on these datasets, and we evaluated the weights’ accuracy through mAP. We set the IoU threshold to 0.5, and the mAP for the nose dataset achieved 76%. This value is comparable to other general object detection accuracy values [22]. The nose was actually detectable when we saw the nose-detected video. In addition, it was well detected regardless of whether one or both nostrils were in the image.

Of the three different analysis methods—maximum temperature, integrating, and averaging—averaging all temperature values in the ROI was found to provide a strong correlation with the actual breathing pattern. For the first method, although the temperature changes depend on the breathing patterns, the maximum ROI temperature showed a range of change that was sometimes small (Figure 5). With the second method, the integration of all temperature values in the ROI, the integrated values sometimes suddenly or gradually increased regardless of the breathing strength (Figure 6 and Figure 7). On the other hand, the results of averaging all the pixel temperature values were the most similar to a normal breathing pattern and were found to be highly correlated with those measured via thermal image observation (R^2^ = 0.91, *p* < 0.001, Figure 8).

## 4. Discussion

This paper overcomes the limitations of previous thermal image research: (1) Some previous infrared research used naked-eye observation to analyze the breathing pattern [5,6]. However, this is labor-intensive. We used an automatic object detection algorithm, making it possible to get a long-time breathing pattern. (2) Other infrared research, not using automatic detection algorithms, used the maximum value to detect the region [8]. This is because the eyes usually show the maximum temperature in the whole body or in the head, so the temperature change in the area can be observed through the maximum temperature. However, chances are high that it will not be representative of temperature values within the ROI. For example, two different breaths cannot be differentiated using only the maximum temperature; shallow breathing accompanies a small region temperature change and deep breathing accompanies a larger region temperature change. In this study, as we used an object detection algorithm, not the maximum temperature, to detect the ROI, we could avoid the problem of using the maximum value and could use all temperature data within the ROI. (3) As we used the Mask R-CNN for ROI detection, we obtained better detection accuracy, and optimal quantization, which was used to maximize the contrast within the thermal image, was no longer necessary. In human breathing pattern research [13], thermal images were used to track the ROI and optimal quantization was used to maximize the detection accuracy. Compared to humans, animals move their body and head much more randomly and actively, so a more accurate detection tool is necessary to apply computer vison methods to animal practice. Mask R-CNN basically uses RGB images, which contain more image characteristics than thermal images, and thus shows greater accuracy. In addition, as the segmentation mask only covered the exact ROI, a more accurate temperature pattern was obtained. Previous research only set the ROI as a square format and unnecessarily included parts that did not correspond to the ROI. Especially in cattle, since the distance between both nostrils is relatively greater than that in humans, chances are high that many unnecessary parts are included together when set in a square shape. This can lead to inaccurate temperature values. (4) In a prior study [13], a more accurate result was obtained when the authors integrated all the temperature values corresponding to the nose region than when they averaged the values. This was because the ROI was detected on thermal images, not RGB images, and the ROI size varied depending on the breathing strength. Thus, the average values between deep breathing and shallow breathing were not different. However, if the integrated method is used for cattle, some problems could appear. First, values can vary depending on the distance between the camera and the object, regardless of the breathing strength. If the distance between the object and the camera is smaller, even if the breathing strength is equal, the number of pixels that the ROI takes up in the image increases and the integrated value increases as well. In practice, we saw an increased value regardless of breathing strength when we brought the camera closer to the cattle or the cattle came closer to the camera (Figure 6). Secondly, the integrated values were different between when there were two nostrils on the image and when there was just one nostril (Figure 7). Even if the breathing strength is equal, when the number of nostrils in the image increases from one to two, it doubles the integrated value. This problem could be critical in cattle because the frequencies of both or one side nostril are more alternating due to their activity. In addition, the fact that the nostrils of cattle are more anatomically three-dimensional than those of humans worsens the problem. On the other hand, using average temperature values, which was problematic in the prior study, did not present problems in our study. This is because we detected the ROI on RGB images and the ROI size was not related to the breathing strength. Thus, we could gain a stable breathing pattern by avoiding the disadvantages of using integral values and solved the problem of using average values in the previous study.

This study was intended to evaluate cattle breathing patterns non-invasively for a long period of time so that fast or shallow breathing patterns could be utilized to detect pain or stress. However, there are some limitations to evaluating other pulmonary diseases, such as pneumonia, and subsequent abnormal pulmonary sounds with this method. We think that auscultation sound is still necessary for these diseases’ diagnosis. In addition, we decided to use the FLIR One Pro because it was most likely to be commercialized considering its price and was able to obtain the sequence temperature data. However, it is lacking in resolution, battery performance, emissivity settings for biological tissue, and accuracy when compared to other infrared cameras. To apply this method in dairy farms and make it a real-time method, it must be developed through further research.

## 5. Conclusions

Breathing patterns are a vital sign that provide health information. Infrared thermography has recently been used to evaluate breathing patterns. We combined an object detection algorithm, Mask R-CNN, with infrared thermography. We recorded RGB images and sequence temperature data simultaneously. The extracted ROI masks from Mask R-CNN were applied to the corresponding temperature data, and we were able to gain stable breathing patterns by averaging the ROI temperature values. The characteristics of this method will make it possible to analyze not only the breathing rate but also diverse breathing information from various aspects.

## Figures and Tables

**Figure 1 animals-11-00207-f001:**
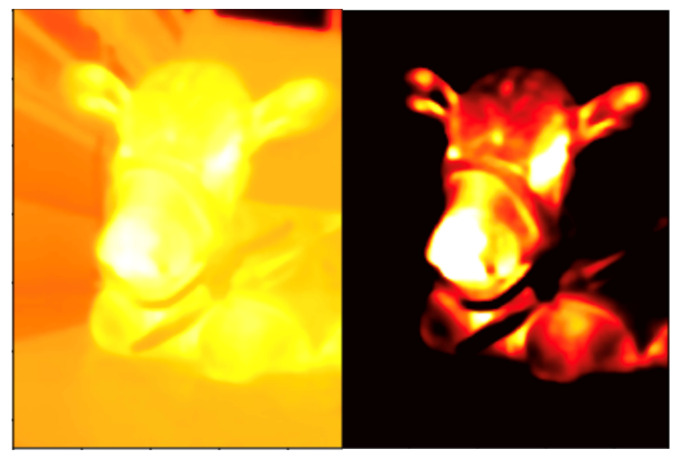
An example of optimal quantization: (**left**) visualization of the temperature values before the optimal quantization; and (**right**) the image after the optimal quantization shows the separation of the object from the background, maximizing thermal image contrast.

**Figure 2 animals-11-00207-f002:**
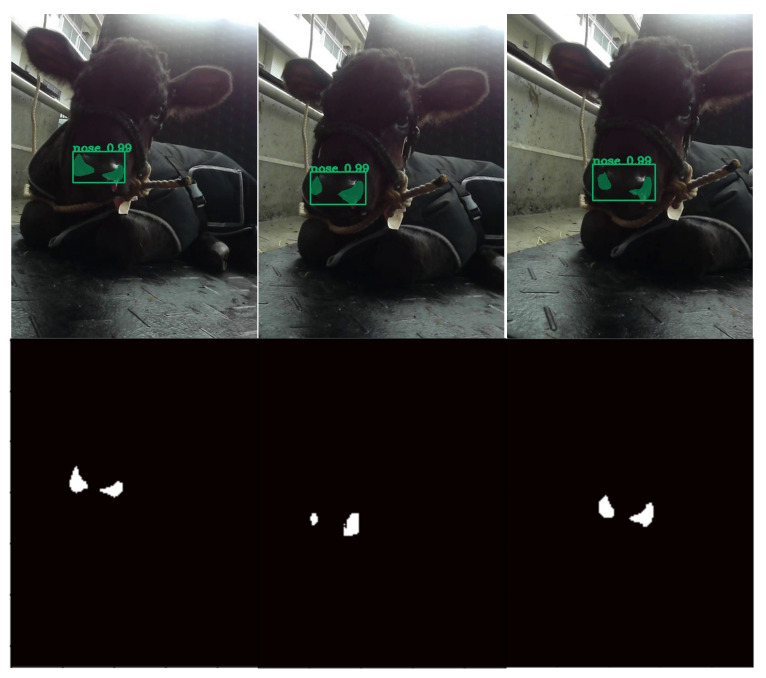
Sample results of nose detection using the proposed method: (**top**) green bounding boxes (detection) and green masks (segmentation) show the ROI (nose) detection; and (**bottom**) the detected masks were extracted for application to the corresponding thermal data.

**Figure 3 animals-11-00207-f003:**
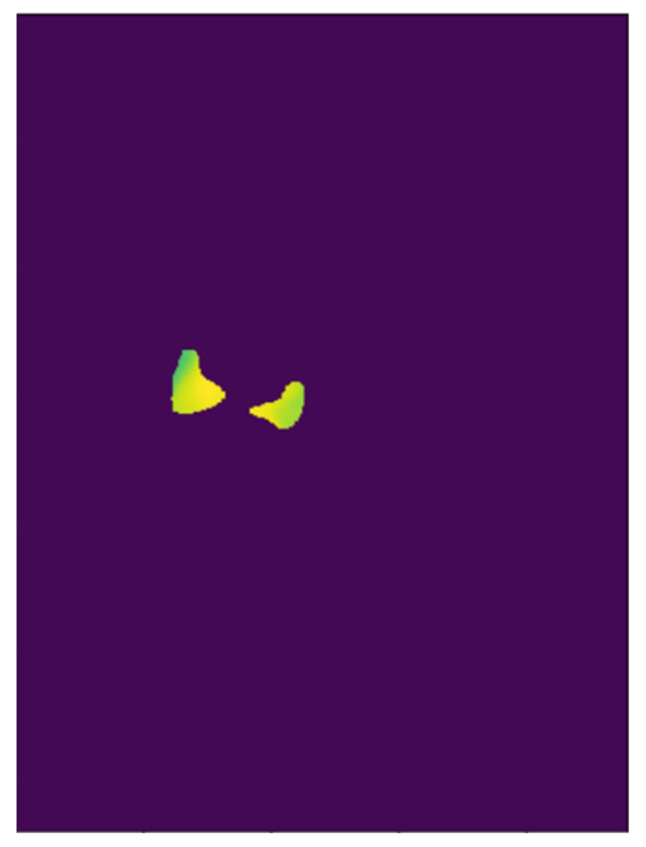
Sample image representing the multiplication of the mask and the corresponding thermal data. On this frame, the average temperature inside the mask was 33.85 °C.

**Figure 4 animals-11-00207-f004:**
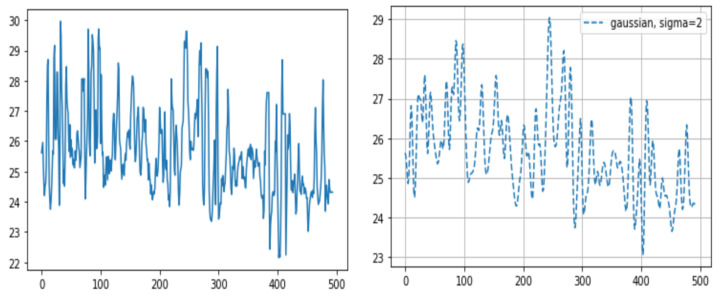
Sample graphs representing the breathing pattern. After multiplication of the masks and the corresponding thermal data through all frames, we were able to gain the breathing pattern. The X-axis represents the frame number (10 fps, frames per second) and the Y-axis represents temperature. (right) The graph after applying the Gaussian window.

**Figure 5 animals-11-00207-f005:**
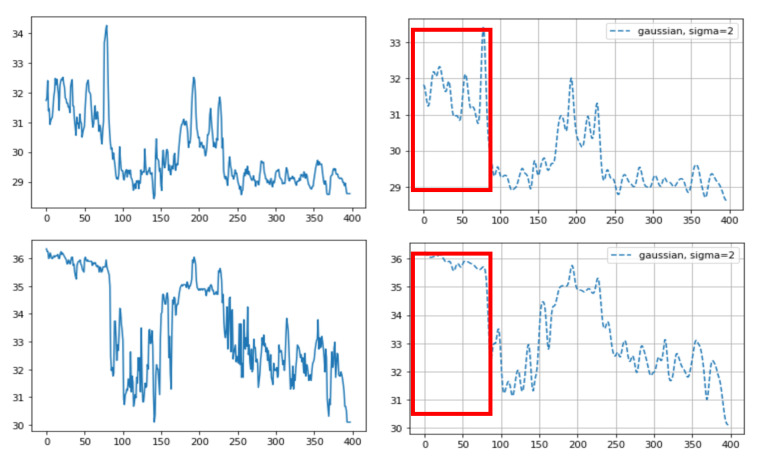
An example of different thermal data calculation methods on the same sample: (**top**) averaging of all the pixel temperature data, before (**left**) and after (**right**) the Gaussian window; and (**bottom**) the maximum temperature in each frame ROI, before (**left**) and after (**right**) the Gaussian window. The temperature fluctuation does not represent the breathing pattern in the case of the maximum temperature method (red box (bottom)). The X-axis represents the frame number (10 fps, frames per second) and the Y-axis represents temperature.

**Figure 6 animals-11-00207-f006:**
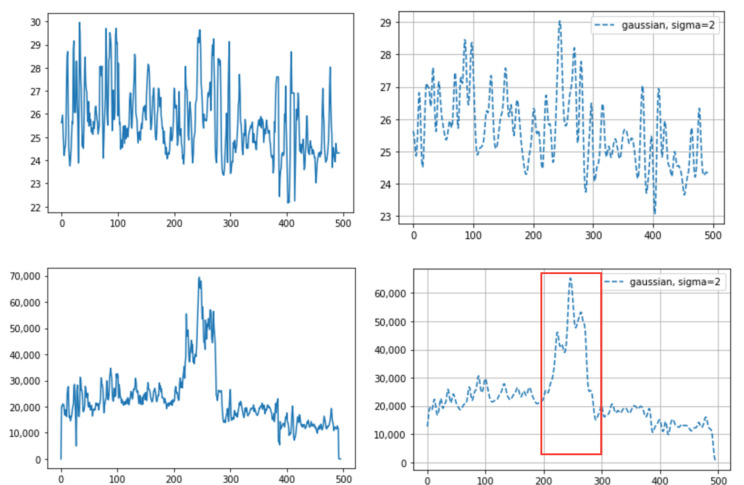
An example of different thermal data calculation methods on the same sample: (**top**) averaging of all the pixel temperature data: before (**left**) and after (**right**) the Gaussian window; and (**bottom**) integrated values of all the pixel temperature data: before (**left**) and after (**right**) the Gaussian window. The integrated values suddenly increased as they approached around the 200th frame (red box (bottom)). In this case, the camera approached the cattle during the time of the 200th frame. The X-axis indicates the frame number (10 fps, frames per second) and the Y-axis indicates temperature (top) or the integrated value (bottom).

**Figure 7 animals-11-00207-f007:**
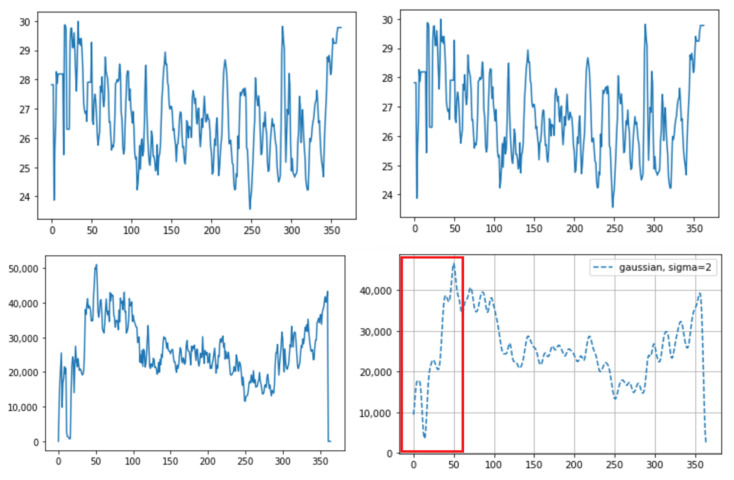
An example of different thermal data calculation methods on the same sample: (**top**) averaging of all the pixel temperature data: before (**left**) and after (**right**) the Gaussian window; and (**bottom**) integrated values of all the pixel temperature data: before (**left**) and after (**right**) the Gaussian window. The integrated values suddenly increased as they approached around the 30th frame (red box (bottom)). This occurred due to the difference between only one nostril detected before the 30th frame and both nostrils detected after the 30th frame due to the cattle head direction. The X-axis represents the frame number (10 fps, frames per second) and the Y-axis represents the temperature (top) or the integrated value (bottom).

**Figure 8 animals-11-00207-f008:**
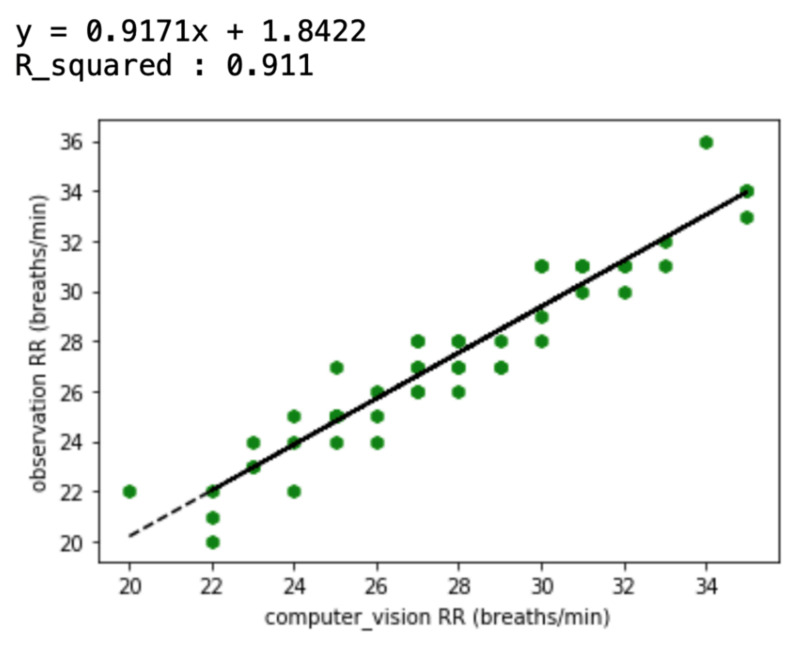
Correlation between respiration rates (RRs) measured via thermal image observation and a computer vision method (averaging all temperature values in the ROI) for 50 recordings.

## Data Availability

No new data were created or analyzed in this study. Data sharing is not applicable to this article.

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
