# Peer review of "Breathing Pattern Analysis in Cattle Using Infrared Thermography and Computer Vision"

_animals, 2021, doi:10.3390/ani11010207_

Round 1
Reviewer 1 Report
The paper is interesting and worthing to be published.
Minor text editing are needed:
line 32:remove period before 1.
line 160:remove period before 3.
line 206:remove period before 4.
line 210:please, check this sentence
line 233: consider if change "varies" with "varied"
line 315: change "ansen" with "Hansen"
line 332: check this reference
Author Response
Response to Reviewer 1 Comments
Point1 : Minor text editing are needed.
Response1 : I revised all the minor text editings you mentioned. In addition, I have completed the English Editing Service provided by mdpi. Please see the attchment files (manuscript and english editing certificate).
Reviewer 2 Report
This paper presents a method of automatic detection of respiration patterns in cattle using computer learned visual detection combined with infrared thermography. The paper requires extensive English language revisions. the paper would be greatly improved by rewriting the methods section to allow replication of the experiment. at present it is unclear.
I would suggest a clearer presentation of the methods to a level that would allow other researchers to replicate this work.
Overall the content is suited to publication but some of the sections are difficult to read due to poor English language.
Author Response
Response to Reviewer 2 Comments
Point 1: The paper requires extensive English language revisions.
Response 1: I have completed the English Editing Service provided by mdpi (please see the certificate attachment file).
Point 2: The paper would be greatly improved by rewriting the methods section to allow replication of the experiment. at present it is unclear. I would suggest a clearer presentation of the methods to a level that would allow other researchers to replicate this work.
Response 2: I detailed the methods to make it more clear.
1) (Line 96) more detailed description about counting the respiratory rate
2) (Line 122) detailed description about the device information (FLIR One Pro)
3) (Line 124) more detailed description of the sequence temperature data
I uploaded the revised manuscript through 'Submit Revised Manuscript' Section, not this 'Author's Reply to the Review Report'.

Reviewer 3 Report
This manuscript describes methodology to detect the region of interest in images of cows to calculate breathing patterns. As potential indicators of welfare continue to be explored, innovative ways to more efficiently measure them are needed. Using infrared thermography to determine changes in temperature as a welfare indicator is laborious, and the use of a detection algorithm to automate this may be helpful to researchers using respiration rate and/or patterns, particularly when working with large amounts of data. I strongly recommend review of the manuscript for English language and clarity.
Author Response
Response to Reviewer 3 Comments
Point 1: I strongly recommend review of the manuscript for English language and clarity.
Response 1: I have completed the English Editing Service provided by mdpi (please see the certificate attachment file). I uploaded the revised manuscript through 'Submit Revised Manuscript' section.

Reviewer 4 Report
Review Breathing Pattern Analysis. Kim and Hadaka
General Comments: The authors present a basic methodology manuscript describing one approach to using a non-invasive infrared thermography technology to monitor respiration rate in a bovine model.
This is a useful contribution to animal physiology. The statistical methods suggested appear to be sound and reasonably accurate and the method would provide researchers and practitioners with a valuable tool for monitoring respiration rate.
I would recommend publication of this manuscript subject to the correction of a few major and several minor corrections as outlined below.
Major Corrections:
- The Authors have used the FLIR One Pro infrared device for their research. FLIR manufacture very good equipment. However, the One Pro is a very basic entry level camera with significant limitations for this type of research.
- The One Pro has a fixed emissivity value of 0.95. The emissivity value used for biological tissue is 0.98. How do the researchers propose adjusting for this difference?
- The researchers report a temperature value to two decimals (L129). The One Pro technology is completely incapable of this level of precision. The researchers need to correct this issue.
- Point 1.2 is important in that to be optimally useful as a diagnostic tool the thermal analysis needs to reflect not just the respiration rate but also the respiratory and lung auscultation sounds (degree of crepitant distress). This would require a much more detailed assessment of the thermal profiles than a mean value. The One Pro technology is not capable of this degree of refinement. The authors at the very least need to acknowledge this.
- Related again to the above points is that the accuracy of the One Pro is ±3C. That degree of accuracy would not likely accommodate a refined assessment of auscultation. It would have been interesting if the authors could have used a higher resolution camera.
- The One Pro has a battery life of approximately one hour. This would be a significant limitation to using the device in any commercial setting such as a dairy.
- Was a stethoscope used as a gold standard to assess the respiration rate?
- L90-91. A data collection time of 5 seconds is probably insufficient for an accurate representation of the respiration rate. How was this time frame validated? This is not a question of the legitimate use of spectral analysis but of practical reference to an accurate respiration rate.
Minor Corrections:
- Throughout the manuscript there are several examples of incorrectly using word tense and sentence structure. It would be recommended that an edit of the English use in the manuscript be completed. Please note this is no criticism of the science.
- L19 – Define ROI here (region of interest?)
- L23 – perhaps change “As” to “to”, with respect to?
- L27 and elsewhere through out the manuscript, a comma before the word “and” is not required.
- L28 – change characteristic to characteristics
- L32 – and elsewhere introduce a space between the number and the word in the titles example “1. Introduction”
- L34 - Here and elsewhere suggest using “cattle” instead of “cow”
- L34 – suggest -through “ the use of or with the aid of”?
- L35 – suggest changing pattern to patterns
- L38 – breathing “characteristics”
- L39 – change have to has
- L42 – change case to cases
- L49 – including
- :50 – humans
- L57 – again change cow to cattle. Also remove the colon (:)
- L59 – change “so far” to “reported to date”?
- L60 – again change cow to cattle
- L61 – perhaps change “is far” to “is greater than in many other species” ?
- L67 “rather than”
- L85 – remove the colon (: )
- L89 change “make” to “makes”
- L90-91 – provide the validation for using five second time period
- :98 – perhaps change “nose part” to “ nose anatomy”
- L101 – change train to training
- L106 “coco” – provide the definition
- L118 – gain – in what?
- L129 – the One Pro is not accurate to two decimals!
- For the Methods in general, please provide the Standard Operating Procedures for the infrared camera including the calibration, sensitivity, precision, accuracy, field of view, distance the images were collected at, if any temperature range was set, justification of the emissivity value and other pertinent technical information.
- For the Results and Discussion please provide a commentary regarding the importance of a auscultation sound or crepitant score system and the possible limitation the One Pro camera may have in this respect.
- Figures – good depiction of using the images and spectral analysis for the purpose intended. Well done.
- References – well referenced
Author Response
Response to Reviewer 4 Comments
Point 1: The Authors have used the FLIR One Pro infrared device for their research. FLIR manufacture very good equipment. However, the One Pro is a very basic entry level camera with significant limitations for this type of research.
Response 1: We checked that the emissivity is fixed at 0.95, and cannot change to 0.98. We agree with your idea that this is the limitation of the FLIR One Pro device (1.1). Plus, ±3C accuracy and battery performance might also be the limitations (1.4, 1.5). Nevertheless, the reason why we used this device is because, first, it was most likely to be commercialized considering its price. Secondly, we were able to obtain the sequence temperature data. We added this limitation of the device and the reason why we used it to the Discussion section (Line 274).
Response 1: (regarding point 1.2) Please see the attachment pdf file. The picture deals with temperature data through Python programme. The original matrix, 46387200, shows 46387200 temperature values in the temperature data file "2020-10-20_14.30.45.dat". Because the size of one frame is 640 * 480 (46387200 =151 * 640 * 480), it means it contains all the temperature data for 151 frames (151 frames are 15 seconds because it takes 10 frames per second). In the picture, only the first 10 temperature values were set. The temperature values were stored in the Kelvin format, which were converted into a Celcius form. Then it can come up to the value to two decimals. I added this to the manusript because it was insufficient in the methodology (Line 124).
Point 2: Was a stethoscope used as a gold standard to assess the respiration rate?
Response 2: Reference 6 (Line 310) shows that visual observation of infrared images are reliable. Based on this, visual observation of infrared images, not stethoscope, was used as gold standard. There was a lack of explanantion, so I added this to the methodology (Line 97).
Point 3: L90-91. A data collection time of 5 seconds is probably insufficient for an accurate representation of the respiration rate. How was this time frame validated? This is not a question of the legitimate use of spectral analysis but of practical reference to an accurate respiration rate.
Response 3: We did not use five seconds of breath, but we measured time for five breaths and calculated respiratory rate with it. This refers to the methodology of reference 6 (Line 310). There was a lack of explanantion, so I added this to the methodology (Line 96).
Point 4: Throughout the manuscript there are several examples of incorrectly using word tense and sentence structure. It would be recommended that an edit of the English use in the manuscript be completed. Please note this is no criticism of the science.
Response 4: I corrected all minor corrections you mentioned (grammar, expressions, sentence stucture and so on). Plus, I have completed the English Editing Service provided by mdpi.
Point 5: For the Methods in general, please provide the Standard Operating Procedures for the infrared camera including the calibration, sensitivity, precision, accuracy, field of view, distance the images were collected at, if any temperature range was set, justification of the emissivity value and other pertinent technical information.
Response 5: I added the information to the methodology (Line 122).
Point 6: For the Results and Discussion please provide a commentary regarding the importance of a auscultation sound or crepitant score system and the possible limitation the One Pro camera may have in this respect.
Response 6: I also added and described the importance of an auscultation and the possible limitation of the One Pro (Line 270).
I uploaded the revised manuscript through 'submit revised manuscript' section.

Round 2
Reviewer 2 Report
The paper has been presented with greatly improved English Language and has had most comments addressed adequately. I still believe methods could be improved but the current standard is suitable for publication.
Author Response
Response to Reviewer 2 Comments
Point 1: I still believe methods could be improved but the current standard is suitable for publication.
Response 1: Instead of elaborating further on the methodology, we thought that it would be great to refer to the references inserted in our manuscript. This is because if we started to describe one by one in more detail in the manuscript, it would be too much. For example, through the line 107 reference, anyone can get detailed instructions on how to use VIA. Plus, in the case of Mask RCNN (line 109), anyone can get detailed method description on the homepage and through the reference. Many people have been using it with the description and we thought that it would be too much to describe the whole method in our paper.
Reviewer 4 Report
The revised manuscript is much improved and the authors have addressed most if not all of the reviewers questions.
The authors have utilized a very creative approach to the use of this technology and are to be commended for their efforts.
Please check L122. The One Pro does not have a resolution of 640 X 480. The resolution is more like 80 X 60. Also, the sensitivity measurements for infrared values are more conventionally expressed in degrees C, not mKelvins.
Please check L127. A note here or elsewhere that the proper emissivity for animal tissue should be 0.98, not 0.95. This is important and would have affected the accuracy of the temperature measurements.
It would have been interesting to see this study conducted with one of the more typical automation cameras in the FLIR series such as an A315, 325 or even an A65 as the higher resolution would have created more accurate thermal measurements, possibly enabling a finer measurement of normal and abnormal respiratory signals.
Author Response
Response to Reviewer 4 Comments
Point 1: Please check L122. The One Pro does not have a resolution of 640 X 480. The resolution is more like 80 X 60. Also, the sensitivity measurements for infrared values are more conventionally expressed in degrees C, not mKelvins.
Response 1: (L122) As you mentioned, we corrected 640 X 480 to 80 X 60, and 70mK to 0.07°C.
Point 2: Please check L127. A note here or elsewhere that the proper emissivity for animal tissue should be 0.98, not 0.95. This is important and would have affected the accuracy of the temperature measurements.
Response 2: (L127) we added the sentence (The proper emissivity for animal tissue should be 0.98, not 0.95. But it was set to 0.95 due to the device's emissivity setting limitation)
Point 3: It would have been interesting to see this study conducted with one of the more typical automation cameras in the FLIR series such as an A315, 325 or even an A65 as the higher resolution would have created more accurate thermal measurements, possibly enabling a finer measurement of normal and abnormal respiratory signals.
Response 3: Thank you for your comment. We agree that we should use the recommended one for a more accurate thermal measurement. We will consider using the devices you recommended in the next step.